# Altered Regulation of the Glucose Transporter GLUT3 in PRDX1 Null Cells Caused Hypersensitivity to Arsenite

**DOI:** 10.3390/cells12232682

**Published:** 2023-11-22

**Authors:** Reem Ali, Abdallah Alhaj Sulaiman, Bushra Memon, Singdhendubala Pradhan, Mashael Algethami, Mustapha Aouida, Gordon McKay, Srinivasan Madhusudan, Essam M. Abdelalim, Dindial Ramotar

**Affiliations:** 1College of Health and Life Sciences, Hamad Bin Khalifa University (HBKU), Qatar Foundation (QF), Education City, Doha 34110, Qatar; reaali@hbku.edu.qa (R.A.); abal36503@hbku.edu.qa (A.A.S.); bnmemon@hbku.edu.qa (B.M.); maouida@hbku.edu.qa (M.A.); emohamed@hbku.edu.qa (E.M.A.); 2Diabetes Research Center, Qatar Biomedical Research Institute (QBRI), Hamad Bin Khalifa University (HBKU), Qatar Foundation (QF), Doha 34110, Qatar; 3Division of Sustainable Development, College of Science and Engineering, Hamad Bin Khalifa University, Doha 34110, Qatar; spradhan@hbku.edu.qa (S.P.); gmckay@hbku.edu.qa (G.M.); 4Nottingham Biodiscovery Institute, School of Medicine, University of Nottingham, University Park, Nottingham NG7 3RD, UK; mashael.algethami@nottingham.ac.uk (M.A.); srinivasan.madhusudan@nottingham.ac.uk (S.M.); 5Department of Oncology, Nottingham University Hospitals, Nottingham NG5 1PB, UK

**Keywords:** SLC2A3, PRDX1, arsenite sensitivity, GLUT3 redox state

## Abstract

Targeting tumour metabolism through glucose transporters is an attractive approach. However, the role these transporters play through interaction with other signalling proteins is not yet defined. The glucose transporter SLC2A3 (GLUT3) is a member of the solute carrier transporter proteins. GLUT3 has a high affinity for D-glucose and regulates glucose uptake in the neurons, as well as other tissues. Herein, we show that GLUT3 is involved in the uptake of arsenite, and its level is regulated by peroxiredoxin 1 (PRDX1). In the absence of PRDX1, GLUT3 mRNA and protein expression levels are low, but they are increased upon arsenite treatment, correlating with an increased uptake of glucose. The downregulation of GLUT3 by siRNA or deletion of the gene by CRISPR cas-9 confers resistance to arsenite. Additionally, the overexpression of GLUT3 sensitises the cells to arsenite. We further show that GLUT3 interacts with PRDX1, and it forms nuclear foci, which are redistributed upon arsenite exposure, as revealed by immunofluorescence analysis. We propose that GLUT3 plays a role in mediating the uptake of arsenite into cells, and its homeostatic and redox states are tightly regulated by PRDX1. As such, GLUT3 and PRDX1 are likely to be novel targets for arsenite-based cancer therapy.

## 1. Introduction

Arsenic is a metalloid found ubiquitously in the environment, and the trivalent (arsenite) and pentavalent (arsenate) are the most abundant forms in the drinking water of many countries, including Argentina, Canada, China, India, Japan, Mexico, the Middle East, Nepal, Poland, Taiwan, Vietnam and the USA, with concentrations varying from 50 to 1000 µg/L [1]. The World Health Organization’s recommended limit is 10 µg/L (10 ppb) [2]. Prolonged exposure to arsenic leads to diseases, including several types of cancers, such as skin, lung, liver and bladder, as well as non-malignant ailments, such as type II diabetes [3]. Both sodium arsenite (NaAsO_2_) and arsenic trioxide (As_2_O_3_) can oxidise proteins as well as damage cellular DNA [4]. Arsenite can bind to proteins exclusively via cysteine residues, interacting with either one, two or three residues, as reported for many proteins, such as hexokinase-2, Zinc finger motifs and RING finger domains [5,6]. Despite the toxic effects of arsenite, it is used for treating acute promyelocytic leukaemia [7]. In human cells, arsenic is believed to enter cells via aquaglyceroporins 7 and 9, and likely by the glucose transporter GLUT1, although the evidence for the latter is tenuous [8]. However, in the yeast *Saccharomyces cerevisiae,* the deletion of seven hexose transporters (Hxt1-7) caused the resulting mutant to be hyper-resistant to arsenite, suggesting that glucose transporters are involved in the uptake of arsenite into the cells [9,10].

Previous work from our laboratory identified the dysregulation of glucose transporter GLUT3 in peroxiredoxin 1 (PRDX1)-deleted cells [11]. PRDX1 is an antioxidant enzyme, which performs two critical functions by serving as (i) an antioxidant to catalyse the decomposition of hydrogen peroxide (H_2_O_2_) and (ii) a chaperone to protect several proteins from oxidative damage [11,12]. For example, oligomeric PRDX1 interacts with p53 and other transcription factors, such as c-Myc and NF-κB, and signalling molecules, including PTEN and AKT, and it protects these proteins from oxidation, thus preventing their degradation [12,13]. In our earlier report (2016), we documented that PRDX1 tightly interacts with the essential DNA repair enzyme APE1, which performs a primary role in the base excision DNA repair pathway, as well as acting as a redox factor [11,14,15,16]. Although PRDX1 can decompose H_2_O_2_, the cells downregulated for PRDX1 did not display sensitivity to this oxidant. Moreover, PRDX1 knockdown cells did not interfere with the APE1 gene or its protein expression levels or the cellular distribution of APE1 or further sensitised APE1 knockdown cells towards H_2_O_2_ [11]. These surprising observations prompted us to examine whether cells downregulated for PRDX1 would display any phenotypes following exposure to other toxic oxidants instead of H_2_O_2_ [11]. While it was a broad undertaking, we sought to narrow the possibilities by performing a preliminary micro-array of PRDX1 knockdown cells and found that at least five metallothionein genes—encoding small cysteine proteins involved in metal detoxification—were downregulated as compared to the control cells. Based on the observations that (i) GLUT3 is dysregulated in PRDX1 knockdown cells and that (ii), in *S. cerevisiae*, several hexose transporters can control the influx of arsenite into the cells, a possibility was raised that PRDX1 could be involved in the molecular process controlling arsenite toxicity via GLUT3.

In human cells, GLUT3 belongs to the glucose transporter gene family of solute carriers 2A (SLC2A), which encodes for the GLUT proteins, which mainly facilitate bidirectional glucose transport across the cell membrane in an ATP-independent manner [17,18,19]. The GLUT family consists of 12 members, and many of them have tissue-specific distributions and different affinities for glucose and other hexose derivatives [20]. The GLUT3 transporter was first cloned from human foetal skeletal muscle, and it shares 64% identity with GLUT1. GLUT3 has a nearly five-fold higher affinity for D-glucose than GLUT1, allowing it to function under restricted glucose levels [20]. It is the main transporter of glucose in neuronal cells [21]. High levels of GLUT3 are associated with an increased incidence of metastasis in several types of cancers, including brain, breast, ovarian and colon, and this is believed to be a result of its high glycolytic efficiency [22,23]. A recent study demonstrated that GLUT3, and not GLUT1, can use its C-terminal to promote invasion by tumour cells, which is independent of its role in glycolytic metabolism [23]. Moreover, in triple-negative breast cancers, GLUT3 expression is associated with poor prognosis and transition from the epithelial to the mesenchymal phenotype [24]. Additionally, GLUT3 can induce an inflammatory tumour micro-environment through induction of the C-X-C Motif Chemokine Ligand 8, which activates macrophages and increases metastasis [25]. Thus, GLUT3 appears to play a role in tumours beyond increasing glucose uptake and metabolic reprogramming, and targeting it might yield therapeutic benefits.

In this present study, we observed that PRDX1 null cells displayed sensitivity to the metalloid arsenite and not to cadmium or H_2_O_2_. We showed that GLUT3 level is increased upon arsenite treatment in PRDX1 null cells, and this correlates with hypersensitivity to arsenite. Cells lacking GLUT3 exhibited resistance to arsenite, and the overexpression of GLUT3 sensitised cells to the metalloid. Moreover, we showed that arsenite uptake into the cells is associated with the level of GLUT3 protein. Based on these findings, we proposed that GLUT3 plays a role in mediating the uptake of arsenite into cells. We further showed that PRDX1 interacts with GLUT3 and protects it from arsenite-induced oxidation. We propose that either the high levels of GLUT3 or low levels of PRDX1 or both can be exploited for arsenite-based cancer therapy.

## 2. Materials and Methods

### 2.1. Compounds and Antibodies

The antibodies used in this study are as follows: anti-PRDX1 clone D5G12, (Cell signalling, Danvers, MA, USA), anti-SLC2A3 (cat. no. TA314539, OriGene, Rockville, MD, USA) and anti β-actin (cat. no. Sc-47778, Santa Cruz Biotechnology, Dallas, TX, USA). Sodium arsenite, cadmium sulphate and MG132 were purchased from Sigma Aldrich, St. Louis, MO, USA. Apigenin was obtained from Cayman Chemical, Ann Arbor, MI, USA.

### 2.2. Cell Lines and Culture

MDA-MB-231 breast cancer cell line and HEK293T kidney embryonic cells were purchased from the American Type Culture Collection (ATCC, Manassas, VA, USA). Cells were cultured in Dulbecco’s Modified Eagle Medium supplemented with 10% FBS and 1% penicillin/streptomycin. Cells were incubated at 37 °C and 5% CO_2_. All cell lines were tested for mycoplasma contamination, and cell lines were authenticated via STR profiling.

### 2.3. GLUT3 Knockdown Using siRNAs

GLUT3 siRNA constructs, as well as negative scrambled control, were purchased from Invitrogen, Inchinnan, UK. Cells were plated in T25 flasks at 70% confluency overnight. The lipofectamine 3000 reagent (Invitrogen, UK) was used to transfect the cells, as per the manufacturer’s protocol. GLUT3 downregulation was evaluated by Western blotting at various time points after transfection (days 3, 5 and 7).

### 2.4. Generation of PRDX1 and GLUT3 Knockouts Using CRISPR/Cas-9 System

gRNAs oligonucleotides targeting PRDX1 or GLUT3 were designed in Snapgene and cloned in the pX330-gRNA vector provided by Feng Zhang (Addgene plasmid # 42230) [26]. Positive clones expressing gRNAs were verified by Sanger sequencing. Cells were transfected with plasmids expressing gRNA using the Fugene HD transfection reagent. At 48 h post-transfection, the editing efficiency was verified using the T7E1 assay. Cells were seeded at low density for single-cell colony isolation. The isolated colonies were verified by Western blotting and Sanger sequencing.

### 2.5. Clonogenic Assays

At least 250 cells per well were seeded overnight in 6-well plates. The plating efficiency for each cell line was pre-determined by plating different cell densities for each line and incubating the cells at 37 °C for 14 days. The plating efficiencies were calculated using the following formula: (number of colonies formed/number of cells incubated) × 100. In order to test arsenite sensitivity, cells were plated overnight in complete media. The following day, freshly diluted arsenite, cadmium or hydrogen peroxide were prepared in Dulbecco phosphate-buffered saline (DPBS); the media were aspirated; and cells were treated with the indicated concentrations in DPBS for 30 min. The compounds were aspirated, and cells were topped up with fresh media and incubated for 14 days to form colonies. After the incubation period, the colonies were washed with PBS, fixed and stained with crystal violet, acetic acid and methanol mixture and counted. The survival fraction (SF) was calculated using the following formula: SF = no. of colonies formed after treatment/no. of cells seeded × plating efficiency. The number of colonies counted was normalised relative to the count of untreated wells, which was considered as 100% survival. In the case of inhibition with Apigenin (20 µM), cells were pre-treated for 24 h, and the next day, the cells were washed, treated with arsenite and monitored using the clonogenic assay.

### 2.6. Cell Proliferation Assays

At least 200 cells/well were seeded in 96-well plates in a complete medium and left to adhere overnight. The following day, cells were treated with fresh arsenite diluted in DPBS for 30 min or left untreated. The supernatants were aspirated, and cells were topped up with fresh culture media and incubated for five days. Cell viability was measured using the MTT cell viability assay (Invitrogen, UK).

### 2.7. RNA Extraction and qRT-PCR Analysis

Cells were plated overnight in T25 flasks and treated with the indicated doses of arsenite in DPBS for 30 min. Then, 3 × 10^6^ cells were collected in the RLT buffer (Qiagen, Hilden, Germany) and subjected to RNA extraction using the RNAeasy mini kit (Qiagen). cDNA was synthesised from the total RNA (0.5 µg) using a high-capacity cDNA reverse transcription kit (Thermofisher Scientific, Waltham, MA, USA), according to the manufacturer’s protocol. Real-time PCR (qPCR) was performed using SYBR Green Master Mix (ThermoFisher Scientific). Samples were run on Quanti Studio 6 Flex qPCR machine. GAPDH was used as a loading control. The primers used were as follows: GLUT3-F, 5′-TGCCTTTGGCACTCTCAACCA; GLUT3-R, 5′-GCCATAGCTCTTCAGACCCAAG; GADPH-F, 5′-GTCTCCTCTGACTTCAACAGCG; GADPH-R, 5′-ACCACCCTGTTGCTGTAGCCAA.

### 2.8. Western Blotting

The protein extracts were prepared by lysing cells in a RIPA buffer (50 mM Tris HCL, 150 mM NaCl, 1% Triton X-100, 0.5% sodium deoxycholate, 0.1% SDS and 1 mM EDTA) supplemented with a protease inhibitor (Sigma Aldrich, St-Loius, MO, USA). Protein quantification was performed using the Pierce micro-BCA kit (Thermofisher Scientific). Samples were run on (4–12%) bis-tris SDS-PAGE gel, and separated proteins were transferred to a 0.22 µm nitrocellulose membrane. Membranes were incubated with primary antibodies (4 °C/overnight). The following day, membranes were probed with HRP-Affinity pure (H + L) secondary antibodies 1:5000 (Jackson Immunoresearch, West Grove, PA, USA) for 1 h at room temperature. Membranes were visualised using Pierce™ ECL Western Blotting Substrate (Thermofisher Scientific). For the non-reducing conditions, in order to detect GLUT3 multimeric forms, extraction was performed using the protocol adapted from Zhang et al. [27]. After drug treatments, the media were aspirated, and cells were incubated with PBS supplemented with 100 mM N-ethylmaleimide (NEM) for 10 min at 4 °C. Then, cells were scraped and pelleted by centrifugation at 6000 RPM for 5 min. Cells were resuspended in lysis buffer (100 mM NaCl, 20 mM Tris-Cl, pH 7.5, 0.5% NP-40, 5 mM CaCl2, 100 mM NEM, 1 x protease inhibitor) on ice for 10 min. Samples were then sonicated and centrifugated at 13,000 rpm for 10 min. The protein extracts were quantified using a micro-BCA assay and prepared for Western blotting in a loading dye with no β–mercaptoethanol. Proteins were separated on 4–12% Tris-glycine gel.

### 2.9. Protein Stability Assays

Cells were plated overnight, then treated with 25 μM MG132 (Sigma) for 3 h to inhibit the proteasome machinery. After MG132 treatment, cells were treated with 100 μM arsenite in DPBS for 30 min or left untreated. Cells were washed with DPBS, collected by trypsinisation, and proteins were extracted for immunoblotting.

### 2.10. Immunoprecipitation

Cells were pelleted and resuspended in a RIPA buffer containing protease inhibitors on ice for 30 min. Lysates were incubated with anti-PRDX1 antibody overnight and then conjugated to protein A magnetic beads for 2 h at room temperature. After IP, the beads were washed thoroughly 4 times with Phosphate buffer saline containing 0.01% Tween 20 and protease inhibitors. The immunoprecipitated proteins were eluted using 4 × SDS loading buffer and then heated at 70 °C for 10 min. Denaturated proteins were separated on 4–12% SDS PAGE gels.

### 2.11. Immunofluorescence

Cells were seeded on the coverslips overnight and then treated with the indicated doses of arsenite in DPBS. Cells were fixed with 4% paraformaldehyde for 30 min and permeabilised with 0.1% triton (Thermofisher Scientific) for another 30 min. After blocking with 3% BSA, cells were incubated with anti-GLUT3 antibody at 4 °C overnight and then labelled with Goat anti-rabbit IgG tetramethylrhodamine (Invitrogen A16129) for 1 h. Slides were prepared in duplicates. The imaging was carried out using a Leica confocal microscope. Nuclear fluorescence quantification was performed in ImageJ software; in the analysis, 100 cells per slide were counted.

### 2.12. GLUT3 EYFP Plasmid Construct

pGLUT3-EYFP-N1 was constructed from the pEYFP-N1 vector containing the Kozak sequence, an initiation methionine and a CMV promoter. Briefly, human GLUT3 was first amplified from the K562 cell cDNA library by PCR using the primers GLUT3- XhoI-F1 (CCGGACTCAGATCTCGAGGCCGCCACCATGGGGACACAGAAGGTCACC) and GLUT3-XbaI-R1 (GTGGCGACCGGTGGATCCCGGACATTGGTGGTGGTCTCCTTAGCAG GC). After verifying the fragment size, the fragment was then subcloned into pEYFP-N1 (NovoProlabs, Shanghai, China, cat. No. V012015) following XhoI and XbaI digestion. Positive clones were verified by Sanger sequencing.

### 2.13. Measuring the Intracellular Accumulation of Arsenite

The intracellular arsenite concentrations were measured by inductively coupled plasma optical emission spectroscopy (ICP-OES). Briefly, cells were plated overnight and then treated the following day with arsenite in PBS for the indicated times and concentrations. Cells were collected by trypsinisation, and pellets were resuspended in non-denaturing lysis buffer (20 mM Tris HCL, 137 mM NaCl, 1% Nonidet P-40 and 2 mM EDTA). The extracts were subjected to sonication at 30% amplitude for 15 s, followed by centrifugation (13,000 rpm for 10 min). The supernatants were transferred to new tubes and digested with 5% nitric acid. The samples were run on Agilent 5110 ICP-OES.

### 2.14. Glucose Uptake Assay

Cells were plated in 96-well plates at 2 × 10^4^ cells/well in complete culture media for 8 h. After adherence to the plates, glucose starvation was performed by incubating the cells in DMEM with no glucose media for 12 h. After starvation, cells were treated with arsenite in a Krebs buffer, and glucose uptake was monitored by using the fluorescent D-glucose analogue (2-NBDG) according to the glucose uptake cell-based assay (Cat. no. 600470; Cayman, Ann Arbor, MI) for the indicated time points. Cells were washed twice with the supplied buffer, and the retained fluorescence was measured with FLUOstar Omega micro-plate reader (BMG Labtech, Ortenberg, Germany) at excitation/emission wavelengths of 485 and 535 nm.

## 3. Results

**PRDX1 null cells are sensitive to arsenite**. Previously, we identified PRDX1 as a protein, which interacts with the DNA repair enzyme APE1 in HeLa cells [11]. Since PRDX1 is known to decompose H_2_O_2_, we assumed that cells deleted for the PRDX1 gene would be sensitive to H_2_O_2_. However, we found that HeLa cells deleted for the PRDX1 gene did not show any more sensitivity to H_2_O_2_ compared to the control cells (Figure 1A,B). We decided to test whether the PRDX1-deleted cells would be sensitive to other agents, which induce the production of reactive oxygen species (ROS). Surprisingly, we found that the PRDX1-deleted HeLa cells were very sensitive to the metalloid sodium arsenite but not to cadmium (Figure 1C and Appendix A). Similar results were obtained with HEK293 (Figure 1D–F) and MDA-MB-231 cells (Figure 1G,H) deleted for the PRDX1 gene. These data indicate that deletion of the PRDX1 gene causes sensitivity to arsenite.

**Arsenite alters GLUT3 expression level in HEK293_PRDX1-deleted cells**. We previously conducted a preliminary micro-array analysis of the HeLa PRDX1-deleted cells and observed that the expression of several genes was altered, including GLUT3 [11]. We decided to focus on the relevance of GLUT3 and evaluated its mRNA expression level using the HEK293 cell line deleted for PRDX1 (Figure 1D). RT-qPCR analysis revealed that GLUT3 mRNA expression was reduced in HEK293 cells deleted for the PRDX1 gene in comparison to the control cells (Figure 2A). However, the arsenite treatment (100 µM for 10 min) significantly increased GLUT3 mRNA level by >4-fold in the HEK293_PRDX1 null cells (Figure 2B). There was no significant change in the control cells (Figure 2B). The induction of the GLUT3 mRNA in the HEK293_PRDX1 null cells correlated with an increased level of the GLUT3 protein, as detected by the anti-GLUT3 antibody (Figure 2C, lane 8 and 9 vs. 7; see quantification in Appendix A). However, the increased level of the GLUT3 protein decreased upon longer exposure (5 µM for 30 min) to arsenite (Figure 2C, lane 12 vs. lane 8). These data suggest that PRDX1 is required to maintain the level of the GLUT3 protein, and in the absence of PRDX1, the arsenite-induced level of GLUT3 appears to be susceptible to time-dependent degradation. No significant induction or loss of GLUT3 was observed in the control cells, except for maintaining a higher basal level of GLUT3 as opposed to the PRDX1 null cells (Figure 2C, lane 1 vs. 7). The GLUT3 mRNA expression level was similarly increased by arsenite in the MDA-MB-231 breast cancer cell line lacking the PRDX1 gene (Figure 2E). Likewise, there was a significant increase in the GLUT3 protein levels in the MDA-MB-231_PRDX1 null cells treated with arsenite as low as 1 µM for 10 min (Figure 2F, lane 8 vs. 7; see quantification in Appendix A). Similar to the HEK293_PRDX1 null cells, the MDA-MB-231_PRDX1 null cells also showed diminishing levels of GLUT3 30 min after its induction with arsenite (Figure 2C, lane 12 vs. 8 and Figure 2F, lane 12 vs. 8, respectively). There was no significant effect of arsenite on the GLUT3 mRNA expression level in the control MDA-MB-231 cells (Figure 2E). We interpret these data to suggest that PRDX1 regulates the level of GLUT3, which is responsible for controlling arsenite toxicity.

**Depletion of GLUT3 confers resistance to arsenite**. In order to directly investigate whether GLUT3 would regulate arsenite toxicity, we downregulated GLUT3 and assessed it for sensitivity towards the metalloid. The HEK293 control cells and PRDX1 null cells were downregulated for GLUT3 using siRNA, and the knockdown efficiency was confirmed by Western blot (Appendix A shown only for the HEK293 control). The knockdown cells were next subjected to a survival assay (Figure 3A). The data revealed that downregulation of GLUT3 caused the HEK293 control cells—as well as the PRDX1 depleted cells—to be resistant to arsenite, but not if these cells were transfected with scrambled control (Figure 3A). Thus, irrespective of the presence or absence of PRDX1, the downregulation of GLUT3 caused resistance to arsenite, suggesting that GLUT3 might independently regulate arsenite toxicity. To further confirm this observation, we deleted the GLUT3 gene in the breast cancer cell line MDA-MB 231 and tested for resistance to arsenite. As expected, these breast cancer cells deleted for GLUT3 were equally resistant to arsenite (Figure 3B,C).

**Expression of GLUT3-EYFP confers sensitivity to arsenite**. In a separate experiment, we designed an expression system using the pEYFP vector bearing the CMV promoter to drive the overexpression of GLUT3 as a tagged GLUT3-EYFP protein in HEK293 cells. These cells were transiently transfected with the pGLUT3-EYFP plasmid and then challenged with arsenite. We observed that these transiently transfected cells displayed hypersensitivity to arsenite, as determined by a proliferation assay, compared to the vector control cells (Figure 3D). This finding supports the notion that GLUT3 could potentially play a direct role in modulating the toxicity of arsenite by regulating its uptake.

**Redistribution of GLUT3-EYFP in response to arsenite**. We exploited the GLUT3-EYFP tagged protein to examine whether GLUT3 would be redistributed in the HEK293 cells following arsenite treatment. Using immunofluorescence analysis, we unexpectedly observed that GLUT3-EYPF was primarily present in the nucleus and appeared as foci (see the red arrows in Figure 3E, upper right panel), while there was no such structure in the vector control. Upon short treatment with arsenite (100 μM for 10 min), we observed a striking redistribution of GLUT3-EYPF within the nucleus and perhaps in the cytoplasm (Figure 3E, lower right panel). This finding suggests that the redistribution of GLUT3 may be involved in triggering the toxicity towards arsenite.

**Indirect immunofluorescence reveals that GLUT3 is degraded in the PRDX1 null cells**. We validated the redistribution of GLUT3 by using an anti-GLUT3 antibody and a different cell line. In the MDA-MB 231 control cells and their PRDX1 null counterparts, GLUT3 staining appeared to be predominantly perinuclear (Figure 4A and 4E, respectively). Upon treatment with arsenite (100 μM for 5, 10 and 15 min), there appeared to be no changes in GLUT3 protein localisation in the control cells (Figure 4B–D vs. Figure 4A and quantified as shown in Figure 4I). However, under the same arsenite treatment conditions, the GLUT3 protein was induced within 5–10 min in the PRDX1 null cells, and the level was increased in the nucleus (Figure 4F,G vs. Figure 4E). Following treatment for 15 min, the GLUT3 protein disappeared within the nucleus, perhaps due to protein turnover (Figure 4H and quantified as shown in Figure 4J). These findings suggest that GLUT3 (i) subcellular distribution is dependent upon PRDX1, and (ii) its homeostatic level is regulated by PRDX1.

**PRDX1 null cells show an elevated level of glucose uptake in response to arsenite as compared to the control**. Since the GLUT3 expression level was stimulated by a low dose of arsenite in the PRDX1 null cells (Figure 2C,F), we checked whether this would correspond to an increase in glucose uptake. Briefly, cells were grown in the absence of glucose in DMEM media for 12 h to deplete the intracellular glucose and then released into the uptake buffer containing fluorescently labelled glucose 2-deoxy-2-[(7-nitro-2,1,3-benzoxadiazol-4-yl)amino]-D-glucose in the absence and presence of arsenite. The data revealed that glucose uptake was nearly constant in the control MDA-MB-231 cells and unimpeded by the presence of arsenite (Figure 5A,B), suggesting that arsenite did not interfere with glucose uptake in these control cells. In contrast, the MDA-MB-231 PRDX1 null cells exhibited a different pattern of glucose uptake, where a low dose of arsenite stimulated glucose uptake, and higher doses decreased the uptake (Figure 5B). The data are consistent with the increase in GLUT3 level in the MDA-MB-231_PRDX1 null cells (Figure 2C, lane 11 vs. 7), and the decreased uptake of glucose is in agreement with the disappearance of GLUT3 (Figure 2C, lane 12 vs. 11). Importantly, these data eliminate the possibility that arsenite is competing for glucose uptake.

**Intracellular accumulation of arsenite in PRDX1 null cells as compared to the control**. Next, we examined whether arsenite would accumulate in the cells using inductively coupled plasma optical emission spectroscopy (ICP-OES). Briefly, cells were grown in DMEM media, and the following day, they were treated with different doses of arsenite. The cells were harvested, washed and monitored for the intracellular accumulation of arsenite. When the MDA-MB-231 control cells were challenged with arsenite (100 µM) for up to 10 min, there was a modest intracellular accumulation of the metalloid (Figure 5C). In contrast, there was a significant intracellular accumulation of arsenite in the MDA-MB-231_PRDX1 null cells treated with a dose of 100 µM for 10 min (Figure 5C). This finding suggests that the stimulated GLUT3 level in the MDA-MB-231_PRDX1 null cells is associated with an increased intracellular accumulation of arsenite.

To investigate whether the related GLUT3 transporter, GLUT1, plays a role in arsenite uptake and sensitivity, we pre-treated the MDA-MB-231 control and the MDA-MB-231_PRDX1 null cells with the GLUT1 inhibitor Apigenin. Cells were then exposed to different doses of arsenite and scored for survivors. The MDA-MB-231_PRDX1 null cells remained sensitive to arsenite compared to the control cells (Figure 5D), indicating that GLUT1 does not appear to mediate the transport of arsenite into the cells.

**GLUT3 interacts with PRDX1 to maintain its stability in response to arsenite**. PRDX1 is well established to also perform a chaperone function by preventing the oxidation and subsequent destruction of several proteins, such as AKT and PTEN, through protein–protein interaction [28,29]. Because of the related phenotype, which exists between GLUT3 and PRDX1, concerning arsenite, we checked whether these two proteins would interact. Using co-immunoprecipitation experiments, we observed that the anti-PRDX1 antibody pulled down the GLUT3 protein, as shown by Western blot analysis (Figure 6A). In addition, we performed the reciprocal experiment and found that the anti-GLUT3 antibody pulled down the PRDX1 protein (Figure 6B). This finding suggests that PRDX1 interacts with GLUT3 to protect it from degradation following arsenite exposure.

Next, we examined whether the decreased level of GLUT3 caused by arsenite exposure in the absence of PRDX1 might be due to proteolysis. In order to perform this, cells were plated overnight and incubated with the proteasomal inhibitor MG132 for 3 h followed by treatment with arsenite (5 µM for 30 min), and the total extract was processed using Western blot analysis (Figure 6C). The arsenite treatment did not affect the level of GLUT3 in the control MDA-MB-231 cells (Figure 6C, lane 3 vs. 1 and Figure 6D showing the quantification), while it decreased GLUT3 level in the PRDX1 null cells (Figure 6C, lane 6 vs. 5 and Figure 6E showing the quantification). However, when the MDA-MB-231_PRDX1 null cells were incubated with MG132 (3 h) followed by treatment with arsenite, there was a substantial accumulation of GLUT3 in the PRDX1 null cells and not in the control cells (Figure 6C, lane 8 vs. lane 6 and Figure 6E showing the quantification). We conclude that in the absence of PRDX1, GLUT3 is susceptible to arsenite-induced modification, leading to its degradation by proteolysis.

**PRDX1 is required to maintain GLUT3 in a multimeric state in response to arsenite**. We examined whether PRDX1 would play a role in regulating GLUT3 structural form in response to arsenite. Briefly, HEK293_WT and HEK293_PRDX1 KO cells were either untreated or treated with 100 µM of arsenite for 15 or 30 min, and the total protein extracts were subjected to PAGE under non-reducing conditions followed by Western blot analysis probed with anti-GLUT3 antibodies (Figure 7).

Under non-reducing conditions and no treatment (UT), GLUT3 exists in various high molecular weight species (>300 kDa) in the HEK293_WT cells (lane 1), while in the HEK293_PRDX1 KO cells, GLUT3 tends to feature in the higher molecular weight forms (lane 6), suggesting that the oxidative environment is responsible for inducing changes to the structural forms of GLUT3. The treatment with arsenite (100 µM for 15 min) caused most of the high molecular weight species of GLUT3 to be converted to a form appearing as a ~300 kDa species in both the WT and the PRDX1 KO cells (lane 2 and lane 7), suggesting that the generation of this ~300 kDa species is independent of PRDX1. It is noteworthy that the H_2_O_2_ treatment (250 µM for 30 min) also produced the same ~300 kDa species as the arsenite treatment (lanes 5 vs. 2 or lanes 10 vs. 7). Since H_2_O_2_ is known to cause the oxidation of proteins, it would appear that the ~300 kDa species generated by arsenite is a result of GLUT3 oxidation. Upon increased dose of arsenite (100 µM for 30 min), the ~300 kDa species of GLUT3 disappeared, followed by the appearance of the high molecular weight species in HEK293_WT (lane 3). In contrast, the increased dose of arsenite (100 µM for 30 min) did not cause the disappearance of the ~300 kDa species, nor the appearance of the high molecular weight form in the HEK293_PRDX1 KO cells (lane 8), suggesting that PRDX1 is required to generate the higher molecular weight species. We propose that arsenite-induced toxicity is dependent upon the continuous presence of the ~300 kDa oxidised species of GLUT3 in the absence of PRDX1.

## 4. Discussion

In this study, we provided evidence that the glucose transporter GLUT3 is involved in the uptake of arsenite, and this unique role is tightly associated with the redox function of the peroxidase PRDX1, which interacts with the transporter. Our findings strongly suggest that high expression levels of GLUT3 or functionally inactive PRDX1 or both will cause cells to be hypersensitive to arsenite and may serve to identify cancer patients who could benefit from arsenite therapy. Previous reports demonstrated the cytotoxic effect of trivalent arsenic derivatives (arsenite, As^III^) in combination with the bis-benzylisoquinoline alkaloid tetrandrine (Tetra) in ER-positive and triple-negative breast cancer cell lines [30,31]. The arsenite and tetrandrine combination triggered apoptotic and autophagic cell death, as well as causing S-phase arrest in MDA-MB-231 breast cancer cells [32]. Indeed, arsenic trioxide (As_2_O_3_) has clinical efficacy in the treatment of relapsed and refractory acute promyelocytic leukaemia (APL) patients [33,34]. These reports raise the possibility of exploiting arsenic compounds for the treatment of breast cancer patients [35]. However, this would require the identification of individual molecules and/or the molecular pathways involved in the response to arsenite, and in our study, we identified PRDX1 and GLUT3 as the key factors.

Arsenite is known to induce ROS-mediated DNA damage in breast cancer cells, stimulate IκB phosphorylation, activate the transcription factor NF-κB and increase the c-Myc and HO-1 protein levels [35]. Since PRDX1 is a sensor of ROS and a regulator of intracellular ROS signalling pathways, the hypersensitivity of PRDX1 null cells to arsenite may be a result of either stress-induced cell death or the accumulation of DNA lesions [36,37]. PRDX1 has been shown to protect several signalling proteins, such as PTEN and AKT, from oxidation-induced inactivation and proteasomal degradation, as well as regulating the signalling of key transcription factors, such as NF-κB, c-Myc and the androgen receptor (AR) [38,39,40]. Thus, the observations that (i) PRDX1 interacts with GLUT3 and that (ii) GLUT3 is unstable in the PRDX1 null cells treated with increasing concentrations of arsenite, but not in the presence of the proteasomal inhibitor MG132, strongly suggest that PRDX1 may play a role in protecting GLUT3 from arsenite-induced oxidation, which would otherwise lead to its degradation. A more tantalising possibility is that the interaction of PRDX1 with GLUT3 might govern GLUT3 molecular state, distribution and function. In such scenario, PRDX1 might minimise the role of GLUT3 in arsenite uptake. Thus, in the absence of PRDX1, this would cause GLUT3 to undergo a structural form poised to actively transport arsenite. In support of this possibility, our immunofluorescence data revealed that GLUT3 level was increased in the nucleus of PRDX1 null cells treated with arsenite (100 µM for at least 5–10 min), while there appeared to be no increase in GLUT3 nuclear level under the same conditions in the control cells. Consistent with the immunofluorescence data, we observed that the overexpression of GLUT3-EYFP from the CMV promoter was also localised to the nucleus and, interestingly, formed nuclear foci in the untreated cells. These GLUT3-EYFP nuclear foci dispersed upon exposure to arsenite, demonstrating that GLUT3 subcellular localisation indeed exerts a role in response to the metalloid. In addition to the fluorescence data, we observed that GLUT3 can undergo molecular changes in response to arsenite, which depend on PRDX1 (see Figure 7). We propose that the cytotoxic effect of arsenite displayed by the PRDX1 null cells could be explained if the oxidised species of GLUT3 directly transports arsenite to the nucleus, where it would inactivate the key components—likely via cysteine oxidation—and prevent nuclear transactions, such as transcription, replication and DNA repair [5].

The notion that GLUT3 possesses the ability to mediate the uptake of arsenite into cells is supported by several key observations. For example, (i) downregulation of GLUT3 or its gene deletion conferred resistance to arsenite upon the cells; (ii) overexpression of GLUT3 sensitised the cells to arsenite; and (iii) cells deleted for the PRDX1 gene showed an increased uptake of arsenite, which corresponded with the increased expression of GLUT3 at both the mRNA and protein levels following exposure to low concentrations of arsenite. The exact mechanism by which GLUT3 mediated arsenite uptake appears to involve its oxidation state; however, this would require more in-depth studies. It is noteworthy that GLUT3 has eight cysteine residues, and any one or multiple residues can conjugate with arsenite and alter GLUT3 structure, such that the dose of arsenite can either activate its transporter function or its interaction with PRDX1 or its stability [3,6]. We are in the process of determining whether arsenite targets specific cysteine(s) of GLUT3 in the presence and absence of PRDX1 and whether the modified cysteine(s) prevents the interaction between GLUT3 and PRDX1 [3,6].

We believed that arsenite uptake may be specific to GLUT3, as inhibition of GLUT1 with the Apigenin inhibitor did not cause the cells to show any more resistance to arsenite as compared to cells untreated with the inhibitor. However, we did not test whether other members of the GLUT family—such as another closely related member, GLUT2—would perform a role in arsenite uptake. It is noteworthy that a recent study identified potential new inhibitors of GLUT3 through in silico library screening [41]. One of the reported small molecule inhibitors, G3iA, can inhibit GLUT3 with an IC50 of ~7 µM. However, G3iA retained some inhibitory activity towards GLUT2 and GLUT1 but at higher IC50 (29 µM) [41]. Thus, there is no GLUT3-specific inhibitor, except for the potential candidate G3iA [41]. It is noteworthy that Glutor is a potent glucose uptake inhibitor, but it targets GLUT1, GLUT2 and GLUT3 simultaneously and has shown anti-tumour activity in vitro in different cancers [42,43,44,45]. In several tumour types, the overexpression of GLUT3 and GLUT1 is exploited by cancer cells as a mechanism to fulfil their high glucose demands [46,47]. In addition, GLUT3 has been shown to play an important role in tumour metastasis and invasion [48,49]. While it seems plausible to use a GLUT3 inhibitor to suppress the growth of cancer cells, it should also be considered that GLUT3 is necessary for the transportation of arsenite into the cells. Based on our findings herein, arsenite monotherapy would be most beneficial for treating cancer patients expressing high levels of GLUT3. The high expression of GLUT3 would trigger the rapid uptake of arsenite to damage the DNA and trigger apoptosis, and the subsequent higher dose of arsenite would inactivate GLUT3 via proteolysis.

In short, PRDX1 deficiency generates metabolic changes in tumour cells, leading to DNA damage and tumour adaptation [50]. Normal levels of PRDX1 have been shown to protect several proteins from oxidative damage, such as the oestrogen receptor, and serve as a factor for favourable prognosis of breast cancers and preventing ROS-induced senescence via p38MAPK [51,52]. Here, we provide evidence that low PRDX1 expression in tumours can be a biomarker for arsenite regimen response. It is expected that arsenite will sensitise tumour cells, which are defective in the PRDX1 function, and that increasing the doses of arsenite could have additional benefits by triggering the degradation of GLUT3, thereby starving the cells of glucose.

## Figures and Tables

**Figure 1 cells-12-02682-f001:**
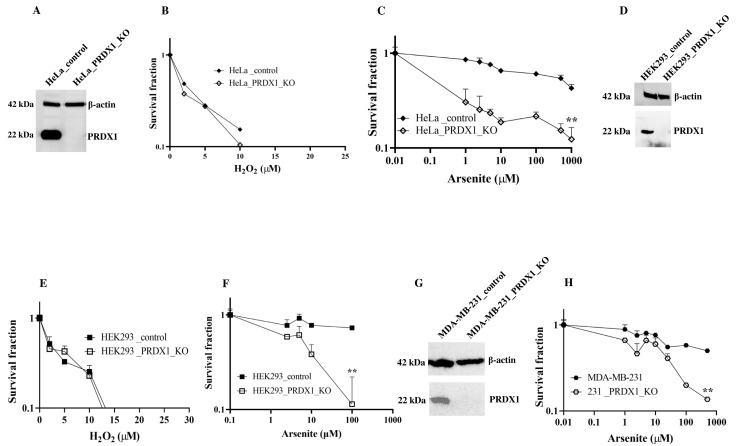
PRDX1 depletion leads to arsenite sensitivity. (**A**,**D**,**G**) Western blot showing control and PRDX1 CRISPR knockout in HeLa, HEK293 and MDA-MB-231 cells, respectively. (**B**,**E**) H_2_O_2_ sensitivity using clonogenic survival assay in control and PRDX1 null HeLa and HEK293 cells, respectively. (**C**,**F**,**H**) Arsenite sensitivity of the control and PRDX1 null HeLa, HEK293 and MDA-MB-231 cells, respectively, using clonogenic survival assay. Survival fraction statistical analysis was performed using a two-way ANOVA test. The error bars represent the mean ± SD ** *p* < 0.01.

**Figure 2 cells-12-02682-f002:**
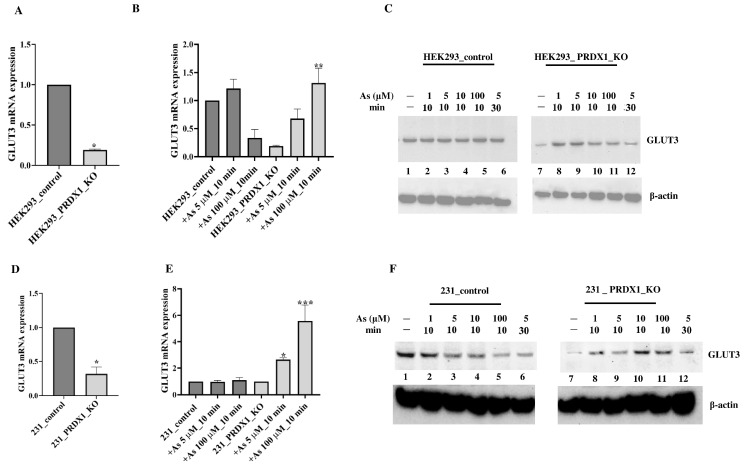
Arsenite upregulates GLUT3 in PRDX1-deleted cells. Briefly, cells were plated overnight in complete culture media; the following day, cells were treated with the indicated doses of arsenite in PBS. After treatment, cells were washed with PBS and collected by trypsinisation, then processed for RNA or protein extraction. (**A**,**D**) GLUT3 mRNA expression levels using RT-qPCR in the indicated control and PRDX1 null cells in the absence of treatment. GADPH was used for normalisation. (**B**,**E**) GLUT3 mRNA expression levels using RT-qPCR in the control and PRDX1 null cells treated with 5 μM and 100 μM arsenite for 10 min. (**C**,**F**) Western blot showing GLUT3 expression levels in the control and PRDX1 null cells treated with the indicated doses of arsenite. β-actin was used as a loading control. Student’s *t*-test was used in (**A**,**D**) to compare GLUT3 mRNA levels in control and PRDX1 knockouts. One-way ANOVA was used in (**B**,**E**). The error bars represent the mean ± SD. * *p* < 0.05, ** *p* < 0.01, *** *p* < 0.001.

**Figure 3 cells-12-02682-f003:**
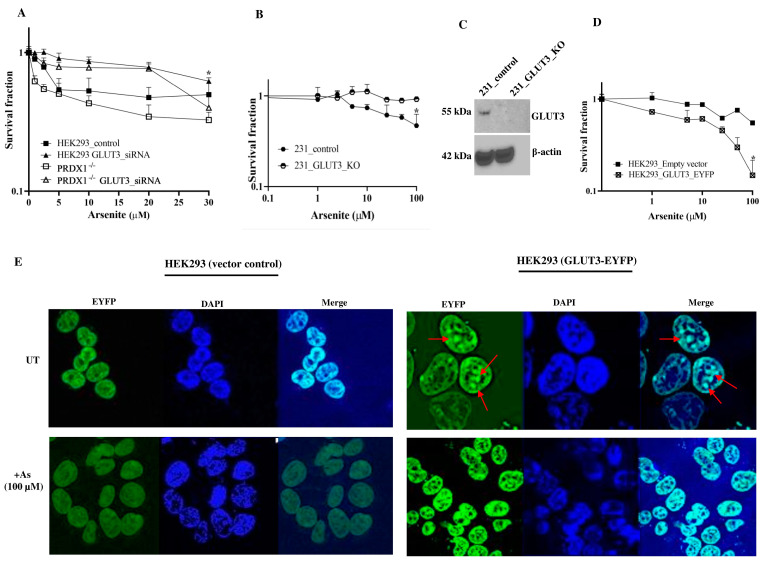
Modulating the GLUT3 level alters the response to arsenite. (**A**) Arsenite sensitivity according to clonogenic survival assay in HEK293 control and PRDX1 knockdown cells transfected with scrambled control or GLUT3 siRNA. (**B**) Clonogenic survival assay showing arsenite sensitivity in the control MDA-MB 231 cells and GLUT3 null cells. (**C**) Western blot showing GLUT3 CRISPR knockout in MDA-MB-231 cells. (**D**) MTT survival assay for arsenite sensitivity in HEK293 cells transfected with the vector control or the plasmid carrying GLUT3 tagged with EYFP. HEK293 cells were plated in 6-well plates overnight and transfected the following day with the vector control or GLUT3-EYFP overexpressing plasmid using the Fugene HD transfection reagent. After 48 h, cells were trypsinised and counted, and 2000 cells/well were re-plated into 96-well plates. After 24 h, cells were treated with the indicated doses of arsenite in PBS for 30 min. The cells were then washed with fresh culture media and incubated in culture media for another 72 h. Cell survival was analysed using the MTT reagent (see the Materials and Methods section). Statistical analysis was performed using two-way ANOVA. The error bars represent the mean ± SD. * *p* < 0.05. (**E**) Representative photomicrographic images of HEK293 transfected with the vector control or with GLUT3-EYFP overexpressing plasmid and either untreated or treated with 100 μM arsenite for 15 min. The magnification used was 63X. The red arrows indicate GLUT3 nuclear foci.

**Figure 4 cells-12-02682-f004:**
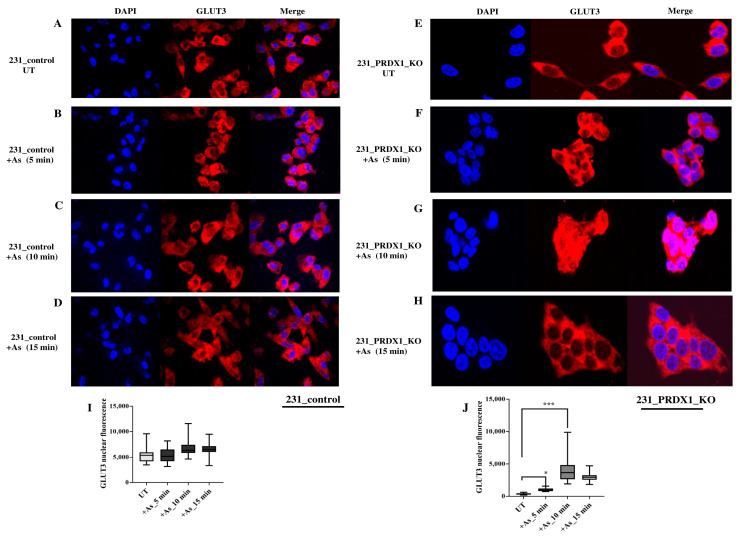
Re-localisation of GLUT3 upon arsenite treatment. (**A**) Representative photomicrographic images showing MDA-MB-231 control cells. (**B**–**D**) showing MDA-MB-231 control cells treated with 100 μM arsenite for 5, 10 and 15 min, respectively. (**E**) Representative photomicrographic images showing MDA-MB-231_PRDX1 null cells. (**F**–**H**) showing MDA-MB-231_PRDX1 null cells treated with 100 μM arsenite for 5, 10 and 15 min, respectively. (**I**,**J**) showing the quantification of GLUT3 nuclear fluorescence by ImageJ software in the control MDA-MB-231 cells and the MDA-MB-231_PRDX1 null cells in the absence and presence of arsenite treatment. Statistical analysis was performed using the one-way ANOVA test. The error bars represent the mean ± SD. * *p* < 0.05 and *** *p* < 0.001.

**Figure 5 cells-12-02682-f005:**
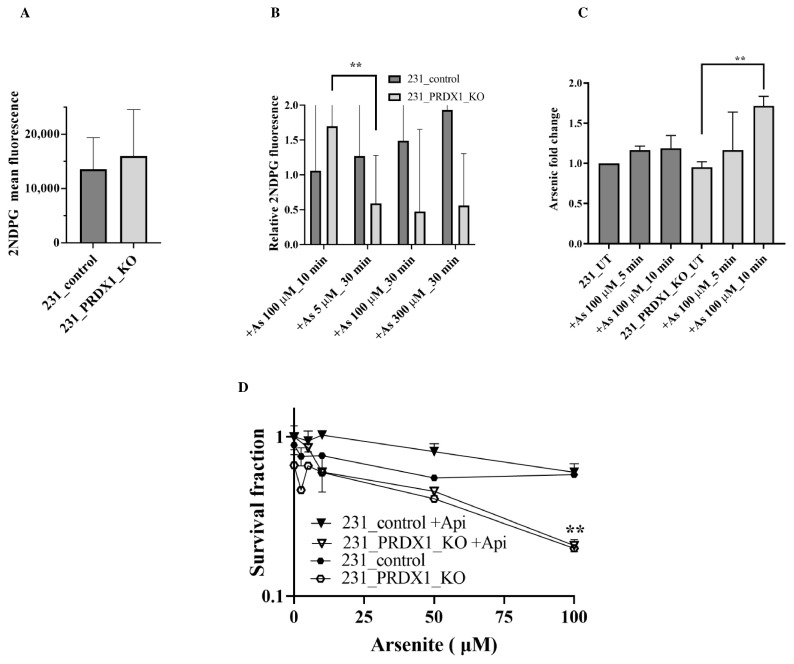
PRDX1 null cells exhibit increased glucose uptake and the accumulation of intracellular arsenic as compared to the control cells. (**A**) Relative glucose uptake in MDA-MB-231_PRDX1 null cells compared to the control cells. (**B**) Relative glucose uptake in MDA-MB-231_PRDX1 null cells treated with the indicated doses of arsenite as compared to the control cells. (**C**) Fold change in the intracellular arsenic level in MDA-MB-231 control and PRDX1 null cells treated with the indicated doses of arsenite. Cells were plated overnight and then treated with the indicated doses of arsenite in PBS. Cells were collected by trypsinisation; then, the pellets were resuspended in a non-denaturing lysis buffer and subjected to sonication to extract intracellular arsenite. Lysates were diluted in 5% nitric acid and analysed on ICP. (**D**) Arsenite sensitivity of the MDA-MB-231 control and MDA-MB-231_PRDX1 null cells pre-treated with the GLUT1 inhibitor Apigenin followed by clonogenic survival assay. Statistical analysis was performed using the one-way ANOVA test. The survival fraction statistical analysis was performed using two-way ANOVA. The error bars represent the mean ± SD. ** *p* < 0.01.

**Figure 6 cells-12-02682-f006:**
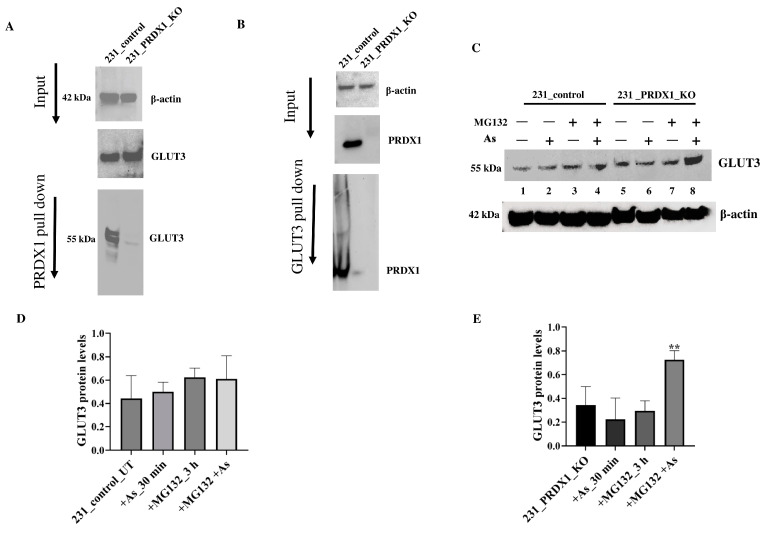
PRDX1 interacts with GLUT3 and protects it from arsenite-mediated degradation. (**A**,**B**) Immunoprecipitation analysis showing the interaction between PRDX1 and GLUT3 in MDA-MB 231 control cells and not in the PRDX1 null cells. (**C**) Western blot showing the GLUT3 protein level in MDA-MB-231 control and PRDX1 null cells following incubation with MG132 and treatment with arsenite. Cells were plated overnight and then treated with the proteasome inhibitor MG132 (25 μM) for 3 h. Cells were then treated with 100 μM arsenite for 30 min or left untreated. Cells were collected by trypsinisation for protein extraction and immunoblotting. (**D**,**E**) Quantification of GLUT3 protein levels by imageJ software in MDA-MB-231 controls and MDA-MB-231_PRDX1 null cells. Statistical analysis was performed using the one-way ANOVA test. The error bars represent the mean ± SD ** *p* < 0.01.

**Figure 7 cells-12-02682-f007:**
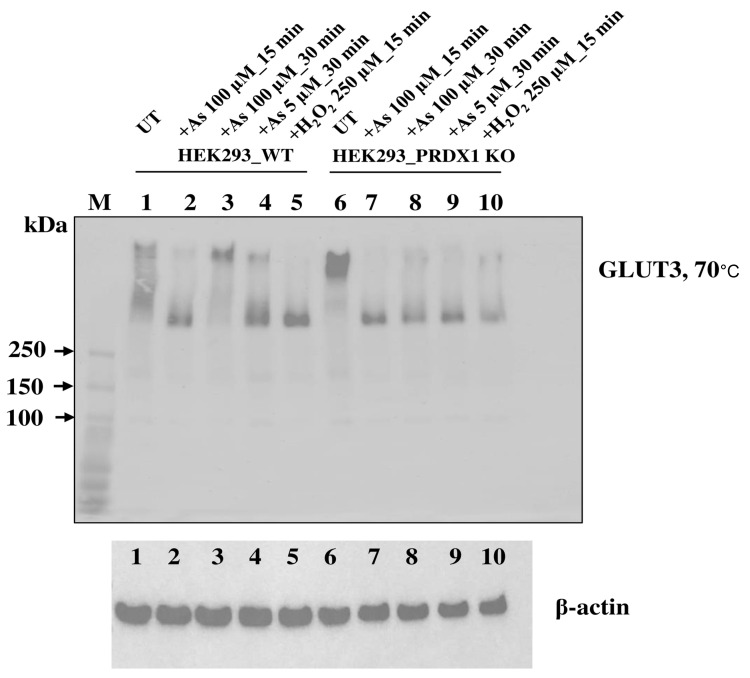
Non-reducing conditions show the multimeric forms of GLUT3 regulated by PRDX1 in response to arsenite. The HEK293_WT cells and HEK293_PRDX1 KO cells were treated with the indicated doses of arsenite or a fixed dose of H_2_O_2_. An amount of 20 μg of total extracts was processed for Western blot analysis under non-reducing conditions using 4–12% tris glycine gels. Lanes 1 and 6, untreated (UT); lanes 2, 3, 7 and 8, treated with 100 µM arsenite for the indicated times (15 or 30 min); and lanes 5 and 10, treated with 250 µM H_2_O_2_ for 15 min. Samples were heated at 70 °C for 5 min before processing on a non-reducing 4–12% gel and probed with anti-GLUT3 antibodies. Anti β–actin was used for assessing the total protein loaded in each well. M, molecular weight standards, kDa.

## Data Availability

The datasets used and/or analysed in the current study are included in the manuscript.

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
