# Peer review of "Altered Regulation of the Glucose Transporter GLUT3 in PRDX1 Null Cells Caused Hypersensitivity to Arsenite"

_cells, 2023, doi:10.3390/cells12232682_

Round 1

Reviewer 1 Report (New Reviewer)

Comments and Suggestions for Authors

The manuscript by Ali et al. reports the involvement of the glucose transporter GLUT3 in arsenite transport. The authors demonstrated that GLUT3 controls arsenite sensitivity in various cell lines depleted of the peroxiredoxin 1, PRDX1. The present study reveals that the GLUT3 mRNA and protein expression levels are responsive to arsenite treatment at short intervals in PRDX1 knock out cell lines. The GLUT3 protein is shown to be induced at 10 min post arsenite treatment, following the disappearance of the signal at 30 min. The authors suggested that this dynamic is due to proteolytic degradation of the GLUT3 protein as confirmed by the treatment of the proteasomal inhibitor MG132. Interestingly, they show a direct interaction between GLUT3 and PRDX1 which they have shown by indirect immunoprecipitation of GLUT3 and the reciprocal IP of PRDX1. The most interesting aspect of this work is that GLUT3 under non reducing conditions exists in various molecular forms and can be directly modified in size by PRDX1 and that certain GLUT3 oxidized forms are involved in the observed arsenite sensitivity.

            The manuscript provides the first evidence that the glucose transporter GLUT3 interacts with PRDX1 that this complex regulates arsenite toxicity. However, I have some minor points that require attention. 

Minor points:

1. In Figure 2, the asterisks representation of the p-value, its not clear what is the comparison here, and which statistical test was used. That should be clarified in the legend. Similar observation is seen in Figure 6, panel E which need to be addressed.

2. The Materials and Methods section describe the construction of pGLUT3-EYFP plasmid for the purpose of this study. The plasmid reference number/sequence should be provided in supplementary materials.

3. The authors should consider in the future to use atomic absorption spectroscopy, if available, which apparently is more sensitive to measure very low concentrations of sodium arsenite in cells.

4. In the future, the authors should continue to investigate which of the Cys in GLUT3 is modified by arsenite, since SNPs might exist to explain why some individuals might be more resistant to arsenite.

Author Response

The manuscript by Ali et al. reports the involvement of the glucose transporter GLUT3 in arsenite transport. The authors demonstrated that GLUT3 controls arsenite sensitivity in various cell lines depleted of the peroxiredoxin 1, PRDX1. The present study reveals that the GLUT3 mRNA and protein expression levels are responsive to arsenite treatment at short intervals in PRDX1 knock-out cell lines. The GLUT3 protein is shown to be induced at 10 min post arsenite treatment, following the disappearance of the signal at 30 min. The authors suggested that this dynamic is due to proteolytic degradation of the GLUT3 protein as confirmed by the treatment of the proteasomal inhibitor MG132. Interestingly, they show a direct interaction between GLUT3 and PRDX1 which they have shown by indirect immunoprecipitation of GLUT3 and the reciprocal IP of PRDX1. The most interesting aspect of this work is that GLUT3 under non-reducing conditions exists in various molecular forms and can be directly modified in size by PRDX1 and that certain GLUT3 oxidized forms are involved in the observed arsenite sensitivity.
The manuscript provides the first evidence that the glucose transporter GLUT3 interacts with PRDX1 and that this complex regulates arsenite toxicity. However, I have some minor points that require attention.
Minor points:
Comment 1: In Figure 2, the asterisk representation of the p-value, it’s not clear what is the comparison here, and which statistical test was used. That should be clarified in the legend. A similar observation is seen in Figure 6, panel E which needs to be addressed.
Response 1: Point well taken. We have now made it clear in the legends and the test being used.
Comment 2: The Materials and Methods section describes the construction of pGLUT3-EYFP plasmid for the purpose of this study. The plasmid reference number/sequence should be provided in supplementary materials.
Response 2: We have added the plasmid and the description as supplementary information.

Comment 3: The authors should consider in the future to use atomic absorption spectroscopy, if available, which apparently is more sensitive to measure very low concentrations of sodium arsenite in cells.
Response 3: The point is well taken, but no laboratory in Qatar possesses this equipment, although there is a plan to acquire this instrument.
Comment 4: In the future, the authors should continue to investigate which of the Cys in GLUT3 is modified by arsenite, since SNPs might exist to explain why some individuals might be more resistant to arsenite.
Response 4: Point well taken. This is certainly the focus of the next phase of the research, especially since arsenite can bind to cysteine residues.

Reviewer 2 Report (New Reviewer)

Comments and Suggestions for Authors

The paper by Reem Ali et al., titled “Altered Regulation of the Glucose Transporter GLUT3 in PRDX1 2 Null Cells Caused Hypersensitivity to Arsenite,” tackles at least two crucial questions. Firstly, it explores the role of 2-Cys peroxiredoxins in cell regulation. Secondly, it delves into optimizing the well-established anticancer activity of arsenic compounds, a topic intertwined with their general toxicity. The research is well-executed. The experimental approach was robust, and the obtained results will be of interest to a wide range of researchers.

I suggest the authors consider adding a picture depicting the growth rate of +/- Prdx1 cells, as it would be beneficial for potential readers.

While the acquired data are robust, the interpretation warrants further refinement. The authors propose that '…GLUT3 directly transports arsenite to the nucleus where it would inactivate key components, likely by cysteine oxidation'. Are the authors certain that As(III) can react with cysteine? I am not aware of such a reaction being possible. Additionally, the explanation for the main finding, 'It is noteworthy that GLUT3 has eight cysteine residues and any one or multiple residues can conjugate with arsenite and alter GLUT3 structure', seems weak. It is highly likely that the eight cysteine residues in GLUT3 exist as disulfides. Although it has been demonstrated that As(III) can associate with Zn fingers, there are no metal ion cofactors in GLUT3.

Author Response

The paper by Reem Ali et al., titled “Altered Regulation of the Glucose Transporter GLUT3 in PRDX1 2 Null Cells Caused Hypersensitivity to Arsenite,” tackles at least two crucial questions. Firstly, it explores the role of 2-Cys peroxiredoxins in cell regulation. Secondly, it delves into optimizing the well-established anticancer activity of arsenic compounds, a topic intertwined with their general toxicity. The research is well-executed. The experimental approach was robust, and the obtained results will be of interest to a wide range of researchers.

Comment 1: I suggest the authors consider adding a picture depicting the growth rate of +/- Prdx1 cells, as it would be beneficial for potential readers.

Response 1: Point well taken.  We have previously done this experiment using HeLa control and HeLa _PRDX1 null cells and found no difference in growth rate.  

Comment 2: While the acquired data are robust, the interpretation warrants further refinement. The authors propose that '…GLUT3 directly transports arsenite to the nucleus where it would inactivate key components, likely by cysteine oxidation. Are the authors certain that As(III) can react with cysteine? I am not aware of such a reaction is possible.

Response 2: Yes indeed arsenite can interact with one, two, and or three cysteine residues of many proteins including hexokinase-2, Zinc finger motifs and Ring finger domains.  We have now highlighted this in the introduction of this revised manuscript with the relevant references. 

Comment 3: Additionally, the explanation for the main finding, 'It is noteworthy that GLUT3 has eight cysteine residues and any one or multiple residues can conjugate with arsenite and alter GLUT3 structure', seems weak. It is highly likely that the eight cysteine residues in GLUT3 exist as disulfides. Although it has been demonstrated that As(III) can associate with Zn fingers, there are no metal ion cofactors in GLUT3.

Response 3: As pointed out above arsenite can bind to one, two or three cysteine residues.  A screen for biotin-As bound proteins using a human proteome microarray with 16,368 GST-affinity purified proteins revealed 360 positives, with enrichment for glycolytic enzymes.  Hexokinase 2 is a key candidate that is inhibited by arsenite.  Arsenite can bind to Cys256, Cys707 and Cys717 of hexokinase 2.  However, these Cys256 and Cys707 are adjacent to the active site and if bound to arsenite would interfere with the enzymatic activity.  So far, arsenite cannot bind to oxidized cysteine as in the disulfide bridge.

Reviewer 3 Report (New Reviewer)

Comments and Suggestions for Authors

The manuscript entitled Altered regulation of the glucose transporter GLUT3 in PRDX1 2 null cells caused hypersensitivity to arsenite seems to contain useful data on PRDX1-GLUT3 interactions that might be of interest to Cells readers, although in its current stage, the aim, work itself and its relevance, does not come across and the paper needs to be improved. The main issue is that the manuscript is not written in form of a story - it feels like the authors jump from one section to another without proper explanation. The introduction is chaotic and does not provide sufficient background on the project, e.g. only in discussion the option on arsenite-based therapy for cancer is mentioned for the first time. The aim is not explained sufficiently. I assume the authors plan to selectively target PRDX1-lacking cancer cells based on their high GLUT3 expression for uptake of arsenite and induced toxicity, however that message does not come across.

The figures should be improved as currently the plots are not well-aligned and multiple sizes of the fonts are used. Additionally, some blots should be rerun as they are overexposed. Figure legends should include the details on statistical analysis. The quality of images in figure 3 needs to be improved, the foci are not visible. In Figure 5D, there should be a comparison between each condition with and without apigenin, as the authors wish to observe the effect of GLUT1 inhibition on arsenite-sensitivity rather than the comparison of arsenite sensitivity between the conditions. It would be helpful if authors included experiments on a healthy cell line to confirm that proposed treatment would be cancer-specific. Additionally the authors should explain in the introduction the reasoning behind choosing arsenite as the cargo for GLUT3 as in the current state that choice seems arbitrary. 

Based on our findings herein, arsenite monotherapy would be most beneficial to treat cancer patients expressing high levels of GLUT3.

Which are? The authors should add the TCGA (or equivalent) analysis of GLUT3/ PRDX1 expression and thus identify potential cancer types that might be sensitive to the proposed regimen.

Comments on the Quality of English Language

Some minor language/syntax errors need to be corrected. The article needs to checked for spelling mistakes.

Author Response

The manuscript entitled Altered regulation of the glucose transporter GLUT3 in PRDX1 2 null cells caused hypersensitivity to arsenite seems to contain useful data on PRDX1-GLUT3 interactions that might be of interest to Cells readers, although in its current stage, the aim, work itself and its relevance, does not come across and the paper needs to be improved.  The main issue is that the manuscript is not written in form of a story - it feels like the authors jump from one section to another without proper explanation.

Comment 1: The introduction is chaotic and does not provide sufficient background on the project, e.g. only in discussion the option on arsenite-based therapy for cancer is mentioned for the first time. The aim is not explained sufficiently. I assume the authors plan to selectively target PRDX1-lacking cancer cells based on their high GLUT3 expression for uptake of arsenite and induced toxicity, however that message does not come across.

Response 1: Sorry and we agreed with the comment.  We have revamped the entire introduction. 

Comment 2: The figures should be improved as currently the plots are not well-aligned and multiple sizes of the fonts are used. Additionally, some blots should be rerun as they are overexposed. Figure legends should include the details on statistical analysis.

Response 2: Point well taken.  We have adjusted all the figures and standardized the fonts.  We included all the details of the statistical analysis.  However, we were unable to redo the β-actin control, but these are included in the raw data file. 

Comment 3: The quality of images in figure 3 needs to be improved, the foci are not visible.

Response 3:  We are not certain why, but in the embedded figures herein, the foci are indeed visible.  We are not sure if the quality was reduced during the reviewing process.   

Comment 4: In Figure 5D, there should be a comparison between each condition with and without apigenin, as the authors wish to observe the effect of GLUT1 inhibition on arsenite-sensitivity rather than the comparison of arsenite sensitivity between the conditions.

Response 4: Point well taken.  The additional controls are now included in Fig. 5D.

Comment 5: It would be helpful if authors included experiments on a healthy cell line to confirm that proposed treatment would be cancer-specific.

Response 5:  Great idea.  Thank you, but we have no comparable primary cell line in our stock and not easy to acquire.

Comment 6: Additionally the authors should explain in the introduction the reasoning behind choosing arsenite as the cargo for GLUT3 as in the current state that choice seems arbitrary. 

Response 6: Thank you, and we have now provided the context of the work. 

Comment 7: Based on our findings herein, arsenite monotherapy would be most beneficial to treat cancer patients expressing high levels of GLUT3. Which are? The authors should add the TCGA (or equivalent) analysis of GLUT3/ PRDX1 expression and thus identify potential cancer types that might be sensitive to the proposed regimen.

Response 7: The point is well taken and we are currently performing TMA analysis with two separate breast cancer cohorts.  We hope from this analysis we can define individuals who have high expression of GLUT3 and low expression of PRDX1.  Cells derived from these patients are expected to be exquisitely sensitive to arsenite.

Comment 8: Some minor language/syntax errors need to be corrected. The article needs to checked for spelling mistakes.

Response 8:  Thank you and we have done our best to correct the grammatical errors.

Round 2

Reviewer 3 Report (New Reviewer)

Comments and Suggestions for Authors

The quality of the manuscript has now greatly improved, but the authors still need to unify the font sizes in Figure 5. I do not raise other issues.

Comments on the Quality of English Language

Minor corrections required. I suggest the manuscript should be read by a native-speaker. 

This manuscript is a resubmission of an earlier submission. The following is a list of the peer review reports and author responses from that submission.

Round 1

Reviewer 1 Report

Comments and Suggestions for Authors

The present manuscript by Ali et al. reports data suggesting that PRDX1 binds to and protects GLUT3 from proteosomal degradation, and that GLUT3 plays a role in the accumulation of arsenic in treated cells.  When the gene for PRDX1 is knocked out, GLUT3 amounts diminish and arsenic is less toxic to cells.  As the authors suggest, this could provide insights that would support checking PRDX1 and/or GLUT3 status in patients before deciding on use of arsenical compounds as chemotherapeutics.

The data presented are suggestive, but more rigor is needed in what is shown and described (as commented on further below).  Most disappointing is that the authors do not adequately mention and in no way address the possibility that PRDX1 may be playing a role in either protecting the GLUT3 protein from oxidation (it has multiple Cys residues), or in causing it to be oxidized under conditions with elevated ROS.  The argument is made that this is solely the chaperone function of PRDX1 that is involved, but no experimental work supports this statement.  If the protein only needs a binding function and not catalytic activity, then cysteine mutagenesis would likely bear this out, but this has not been tried.

Major

The minimal mention of protein oxidation in the manuscript particularly neglects the role that PRDX proteins play in “redox relays” through which they can mediate the oxidation of proteins to which they bind.  Cited literature refers more to protein oxidation rather than just the “stability” of the proteins to which PRDX1 binds.

The authors “assumed” PRDX1 deletion would affect H2O2 scavenging but the bolus challenge to cells did not bear that out.  Unmentioned and not explored experimentally is that other PRDX proteins (2-6) may be upregulated in surviving cells.  One would expect to see experimental data addressing this issue.

The MTT (not MTS as in Fig. 3D legend) assay is used here as a measure of cells and their survival, but is not a good choice (meant to be used to monitoring proliferation, as the name of the assay indicates), particularly when a toxin or other perturbant is used that provokes ROS generation given the redox sensitivity.

I am having a hard time seeing in the images of Figures 3 and 4 what is described in the text.  The authors describe movement of the GLUT3-EYFP from nucleus to cytosol with AS, but in Fig. 3E 3 out of the 4 panels seem to have relatively poor/different DAPI staining that is more apparent, not the other panels.  The red cell images on the right side of Fig. 4 (PRDX1 KO) are all very indistinct as far as showing individual cells.  Exactly how quantitation is done, particularly with some of these densely plated cells, is not clear; it seems to be “nuclear fluorescence” (Fig. 4) but not on a per cell basis.  These aspects could be greatly improved.

Fig. 5 and the associated text is also of concern.  The most strange aspect is that in Fig. 5C, AS amounts from ICP-OES are given as “fold change”, and with incubation with 100 uM AS, none but the 10 min treatment of the KO are different from control.  Why would untreated control cells have ANY arsenic at all other than the tiniest residual amounts?  This is very hard to make sense of, I would expect a huge “fold change”.  Further, arsenic is added and cell-associated amounts change over time, so “exhibit elevated level of arsenite” is an odd way to describe this, “uptake” or “accumulation” makes more sense.  The stated result in lines 361-363 is not at all what is shown in Fig. 5C (that it goes down again at 10 min).

The IP data of Fig. 6 is minimal and should be done more rigorously.  Why don’t we see the GLUT3 present in the input, and not just after pulldown?  (“Input” only shows beta-actin).  And the usual rigor with such experiments is to also do the reverse IP (in case of antibody non-specificity), so GLUT3 should be pulled down followed by blotting for PRDX1.

Minor

There is no meaning to “% homology” (on lines 47 and 52); it should be reported as “% identity”, and would be referring to amino acid sequence.

There are some typos and problems in the description of Fig. 2 results; on line 248 Fig. 2C should be 2F, and 1 mM should be 1 micromolar.  The statement (248-250) about differences at 10 and 30 min after 5 uM AS is not at all obvious with the single gel shown (2F); the quantitation of multiple gels is needed here, not just in the supplement.  (The supplement shows panels of a figure, but has no figure legend.)  The beta-actin used as a loading control is quite overexposed in these gels.  Fig. 4H on line 325 should be 4I.

Teleological language like that used in lines 387-388 is best avoided in scientific writing.

There are multiple places with problematic sentences or non-sentences (e.g. 61, 503) and mistakes, for example right in the title.  Plurals are sometimes missed; note that the word “data” is plural.  On lines 36 and 37, “encode for” should either be “encode” or “code for”. A reference is needed for the sentence ending on line 65.  Reference 32 shown on line 506 does not seem to match the statement.

Comments on the Quality of English Language

Noted above (several non-sentences, some mistakes and typos).  Mostly adequate.

Author Response

Reviewer 1

Comments and Suggestions for Authors

The present manuscript by Ali et al. reports data suggesting that PRDX1 binds to and protects GLUT3 from proteosomal degradation, and that GLUT3 plays a role in the accumulation of arsenic in treated cells. When the gene for PRDX1 is knocked out, GLUT3 amounts diminish and arsenic is less toxic to cells. As the authors suggest, this could provide insights that would support checking PRDX1 and/or GLUT3 status in patients before deciding on use of arsenical compounds as chemotherapeutics.

The data presented are suggestive, but more rigor is needed in what is shown and described (as commented on further below). Most disappointing is that the authors do not adequately mention and in no way address the possibility that PRDX1 may be playing a role in either protecting the GLUT3 protein from oxidation (it has multiple Cys residues), or in causing it to be oxidized under conditions with elevated ROS. The argument is made that this is solely the chaperone function of PRDX1 that is involved, but no experimental work supports this statement. If the protein only needs a binding function and not catalytic activity, then cysteine mutagenesis would likely bear this out, but this has not been tried.

Major

Comment 1: The minimal mention of protein oxidation in the manuscript particularly neglects the role that PRDX proteins play in “redox relays” through which they can mediate the oxidation of proteins to which they bind. Cited literature refers more to protein oxidation rather than just the “stability” of the proteins to which PRDX1 binds.

Response 1: This point is well taken and we have now discussed that PRDX1 interaction with GLUT3 is to protect the cysteine residues of GLUT3 from As-induced modification (see discussion pg 23, lines 16-22).  We also mentioned that we are in the process of isolating GLUT3 to check which of the cysteine(s) residue is modified by arsenite in the presence and absence of PRDX1 (see discussion pg 23, lines 20-22).

Comment 2: The authors “assumed” PRDX1 deletion would affect H2O2 scavenging but the bolus challenge to cells did not bear that out. Unmentioned and not explored experimentally is that other PRDX proteins (2-6) may be upregulated in surviving cells. One would expect to see experimental data addressing this issue.

Response 2: In 2016, we identified PRDX1 as an interacting partner with the DNA repair protein APE1 (see Ref 17).  While downregulation of APE1 sensitized HeLa cells to H2O2, this was not the case for PRDX1, and PRDX1 downregulation did not further sensitize APE1 downregulated cells to H2O2.  This prompted us to check by microarray whether other genes would be upregulated in the PRDX1 deleted cells to compensate for PRDX1 depletion.  We found no indication that other members of the PRDX family were upregulated.  Since the microarray data may not be representative of all genes, we plan to recheck this by performing a more detailed gene expression analysis using RNAseq from HEK293 control and HEK293-PRDX1 null cells in the absence and presence of H2O2.  More importantly, the RNAseq will be conducted with arsenite treatment to examine the differentially expressed gene in the absence of PRDX1.

Comment 3: The MTT (not MTS as in Fig. 3D legend) assay is used here as a measure of cells and their survival, but is not a good choice (meant to be used to monitoring proliferation, as the name of the assay indicates), particularly when a toxin or other perturbant is used that provokes ROS generation given the redox sensitivity.

Response 3: We have corrected the MTS to MTT.  Please note that all the killing curves were obtained by clonogenic survival assay, except for the data shown in Fig. 3D, which was performed by MTT assay as these cells were transiently transfected with the pGLUT3-EYFP expression plasmid.  We did not select these transiently transfected cells for stable clones to carry out clonogenic assays. 

Comment 4: I am having a hard time seeing in the images of Figures 3 and 4 what is described in the text. The authors describe movement of the GLUT3-EYFP from nucleus to cytosol with AS, but in Fig. 3E3 out of the 4 panels seem to have relatively poor/different DAPI staining that is more apparent, not the other panels.

The red cell images on the right side of Fig. 4 (PRDX1 KO) are all very indistinct as far as showing individual cells. Exactly how quantitation is done, particularly with some of these densely plated cells, is not clear; it seems to be “nuclear fluorescence”(Fig. 4) but not on a per cell basis. These aspects could be greatly improved.

Response 4: Points are well taken.  For Fig. 3, the DAPI staining for the three panels is now properly adjusted. For Fig. 4, we have independently repeated the experiment and obtained better images.  We rewrote the section highlighting that arsenite-induced increase level of GLUT3 in the nucleus, which subsequently disappeared.   

Comment 5: Fig. 5 and the associated text is also of concern. The most strange aspect is that in Fig. 5C, AS amounts from ICP-OES are given as “fold change”, and with incubation with 100 uM AS, none but the 10 min treatment of the KO are different from control. Why would untreated control cells have ANY arsenic at all other than the tiniest residual amounts?  This is very hard to make sense of, I would expect a huge “fold change”. Further, arsenic is added and cell-associated amounts change over time, so “exhibit elevated level of arsenite” is an odd way to describe this, “uptake” or “accumulation” makes more sense. The stated result in lines 361-363 is not at all what is shown in Fig. 5C (that it goes down again at 10 min).

Response 5: We consulted with the expert chemist (Dr. S. Pradhan) who conducted the experiment and she stated that the ICP-OES instrument does generate a background noise.  We could have easily removed the noise, but she claimed this is how they have represented all their published data from the instrument.  We now described the data as the accumulation of arsenite rather than uptake (see pg 18, lines 17-26).   The stated result “that it goes down again at 10 mins” in the original manuscript line 363 is an error. This is now corrected.

Comment 6: The IP data of Fig. 6 is minimal and should be done more rigorously. Why don’t we see the GLUT3 present in the input, and not just after pulldown? (“Input” only shows beta-actin). And the usual rigor with such experiments is to also do the reverse IP (in case of antibody non-specificity), so GLUT3 should be pulled down followed by blotting for PRDX1.

Response 6: This is now done and a new panel Fig. 6B is included to show the reciprocal pulldown experiment whereby GLUT3 pulled down PRDX1.  Both Fig. 6A and 6B now showed GLUT3 and PRDX1 input, respectively.

Minor

Comment 1: There is no meaning to “% homology” (on lines 47 and 52); it should be reported as “% identity”, and would be referring to amino acid sequence.

Response 1: Agreed and this is now changed to % identity.

Comment 2: There are some typos and problems in the description of Fig. 2 results; on line 248 Fig. 2C should be 2F, and 1 mM should be 1micromolar.

Response 2: The errors and typos are corrected.

Comment 3: The statement (248-250) about differences at 10 and 30 min after 5 uM AS is not at all obvious with the single gel shown (2F); the quantitation of multiple gels is needed here, not just in the supplement. (The supplement shows panels of a figure, but has no figure legend.)

Response 3: The data are derived from multiple gels and for simplicity, we kept the quantification as supplementary data (Fig. S1).  There was a legend for Fig. S1 and it was at the end of the Figure legends section (see pg 32 lines 12-18). 

Comment 4: The beta-actin used as a loading control is quite overexposed in these gels. Fig. 4H online 325 should be 4I.

Response 4: This occurred as we kept the same time of exposure for both antibodies.  We agreed we could have done different time exposure.  In the new images for Fig. 4, we decided to eliminate the quantification (4I), as it is very difficult to have an unbiased analysis even with a different confocal microscope.

Comment 5: Teleological language like that used in lines 387-388 is best avoided in scientific writing.

Response 5: We could not find the exact line numbering, but did our best to eliminate the teleological language.

Comment 6: There are multiple places with problematic sentences or non-sentences (e.g. 61, 503) and mistakes, for example right in the title. Plurals are sometimes missed; note that the word “data” is plural. On lines 36 and 37, “encode for” should either be“encode” or “code for”.

Response 6: Totally agreed and these have been corrected as well as the English language to the best of our knowledge.

Comment 7: A reference is needed for the sentence ending on line 65. Reference 32 shown on line 506 does not seem to match the statement.

Response 7: The reference is now added, and Ref 32 is corrected.

Comment 8: Comments on the Quality of English Language

Noted above (several non-sentences, some mistakes and  typos). Mostly adequate.

Response 8: Points are well taken and we have done our best to correct the English grammar throughout the text.

Submission Date 11 May 2023

Date of this review

01 Jun 2023 23:45:02 Author Report Rating (Optional)

Reviewer 2 Report

Comments and Suggestions for Authors

The paper reports the modulation of GLUT3 expression/location and, consequently, function by PRDX1. As demonstrated here, beside a high affinity to glucose, GLUT3 seems also to contribute to arsenite entrance. The conclusions are consistent with the results, the paper is well written and I have only one comment, criticism:

-          The text on the line  271 says that both wild-type and PRDX1 knockout HEK293 had GLTU3 silenced by siRNA. However, the suppl figure 1 shows only the wild-type one.

Author Response

Reviewer 2

Comments and Suggestions for Authors

The paper reports the modulation of GLUT3 expression/location and, consequently, function by PRDX1. As demonstrated here, besides a high affinity to glucose, GLUT3 seems also to contribute to arsenite entrance. The conclusions are consistent with the results, the paper is well written and I have only one comment, criticism:

Comment 1- The text on the line 271 says that both wild-type and PRDX1 knockout HEK293 had GLTU3 silenced by siRNA. However, the suppl figure 1 shows only the wild-type one.

Response 1: Agreed, and we have corrected the statement and wrote “Supplementary Fig. S1F shown only for the HEK293 control”. We have subsequently made deletion of PRDX1 in all the cell lines so the siRNA became redundant.

Round 2

Reviewer 1 Report

Comments and Suggestions for Authors

Authors have responded in part to the points made in my previous review, and a number of things have been fixed or at least improved.  However, the negative aspects that remain are still sufficient for rejection of the paper.  Needed from the authors is a deeper consideration of how the work could be improved, and further experimental work is strongly recommended to improve aspects that were brought up in the previous review.

As minor comments, looking through the figures carefully, none of the cell images show a scale bar.  In addition, the “*” on figures is defined in terms of p-value in legends, but often it is not clear what is being compared, e.g. when the * sits on top of one of the bars, or to the right of a line in a line graph.

A major issue with the previous manuscript was the near total absence of mentioning the actual catalytic activity of peroxiredoxins, which is to reduce hydroperoxides, and the lack of description of another aspect of PRDX function in modulating signaling as far as their ability to serve as mediators of protein oxidation (disulfide bond formation) with their binding partners (commonly termed redox relays).  There has been a very modest attempt to add a mention of PRDX activity as playing a role in the GLUT3 stabilization, but this does not come until the discussion, no data addresses it, and the mention of “functionally inactive PDRX1” (line 393) is a stretch (note the misspelling of PRDX1, which is also still present in the TITLE in spite of this having been mentioned in the previous review).  As described now, the authors equate chaperone activity to thiol protection (in the response to the reviewer comment and in the manuscript), which is now conflating two aspects.  Some but not all PRDXs have chaperone activity, and for a subset of these, the redox state of the protein and presence or absence of the catalytic cysteines may matter (but not always).  The chaperone activity is as a “holdase”, which refers to binding of unfolded or partially unfolded proteins to prevent their aggregation (without a dependence on ATP).  The (1) catalytic or (2) redox relay activity, which could either (1) protect from H2O2-mediated thiol oxidation in the target protein or alternatively (2) cause thiol oxidation in the target protein, may also occur, and may or may not impact “chaperone” activity.  “Stability” is yet another phenomenon that is taken here to apparently mean susceptibility to proteosomal degradation, but this is also loose terminology.  So the comments as they stand now serve to make these aspects MORE murky.  Because mechanisms matter, I do not find the efforts to respond to this criticism to be at all adequate.  Especially given that PRDX1 “function” (which one?) is suggested to be what is important, more besides just a knockout should have been done to investigate what form of the PRDX1 in fact imparts its effect on GLUT3 “stability”.  Overall, my view is that the responses to major criticisms 1-3 have been inadequate.